# Marrying Pixel and Latent Diffusion Models for Text-to-Video Generation

*Close up of mystic cat, like a buring phoenix, red and black colors.*

*A panda besides the waterfall is holding a sign that says "ICLR".*

*Toad practicing karate.*

*Giant octopus invades new york city.*

*A medieval knight in gleaming armor rides confidently on a majestic horse, the sun setting behind them.*

*Motorcyclist with space suit riding on moon with storm and galaxy in background.*

Figure 1: Given text description, our approach generates highly faithful and photorealistic videos. *Click the image to play the video clips. Best viewed with Adobe Acrobat Reader.*

## Abstract

Significant advancements have been achieved in the realm of large-scale pre-trained text-to-video Diffusion Models (VDMs). However, previous methods either rely solely on pixel-based VDMs, which come with high computational costs, or on latent-based VDMs, which often struggle with precise text-video alignment. In this paper, we are the first to propose a hybrid model, dubbed as MPL-Video , which marries pixel-based and latent-based VDMs for text-to-video generation. Our model first uses pixel-based VDMs to produce a low-resolution video of strong text-video correlation. After that, we propose a novel expert translation method that employs the latent-based VDMs to further upsample the low-resolution video to high resolution. Compared to latent VDMs, MPL-Video can produce high-quality videos of precise text-video alignment; Compared to pixel VDMs, MPL-Video is much more efficient (GPU memory usage during inference is 15G vs 72G). We also validate our model on standard video generation benchmarks. Our code will be publicly available and more videos can be found here.

# 1 Introduction

Remarkable progress has been made in developing large-scale pre-trained text-to-Video Diffusion Models (VDMs), including closed-source ones (*e.g.*, Make-A-Video (Singer et al., 2022), Imagen Video (Ho et al., 2022a), Video LDM (Blattmann et al., 2023a), Gen-2 (Esser et al., 2023)) and open-sourced ones (*e.g.*, VideoCrafter (He et al., 2022), ModelScopeT2V (Wang et al., 2023a). These VDMs can be classified into two types: (1) Pixel-based VDMs that directly denoise pixel values,

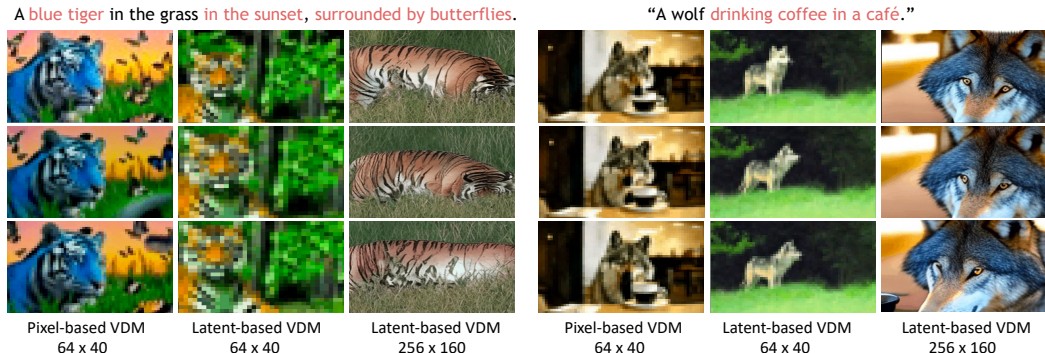

Figure 2: Text-Video alignment comparisons among pixel-based VDM at low resolution, latent-based VDM at low resolution and latent-based at relatively high resolution.

including Make-A-Video (Singer et al., 2022), Imagen Video (Ho et al., 2022a), PYoCo (Ge et al., 2023), and (2) Latent-based VDMs that manipulate the compacted latent space within a variational autoencoder (VAE), like Video LDM (Blattmann et al., 2023a) and MagicVideo (Zhou et al., 2022).

However, both of them have pros and cons. **Pixel-based VDMs** can generate motion accurately aligned with the textual prompt but typically demand expensive computational costs in terms of time and GPU memory, especially when generating high-resolution videos. **Latent-based VDMs** are more resource-efficient because they work in a reduced-dimension latent space. But it is challenging for such small latent space (*e.g.*, $8 \times 5$ for $64 \times 40$ videos) to cover rich yet necessary visual semantic details as described by the textual prompt. Therefore, as shown in Fig. 2, the generated videos often are not well-aligned with the textual prompts. On the other hand, if the generated videos are of relatively high resolution (*e.g.*, $256 \times 160$ videos), the latent model will focus more on spatial appearance but may also ignore the text-video alignment.

To marry the strength and alleviate the weakness of pixel-based and latent-based VDMs, we introduce MPL-Video, an efficient text-to-video model that generates videos of not only decent video-text alignment but also high visual quality. Further, MPL-Video can be trained on large-scale datasets with manageable computational costs. Specifically, we follow the conventional coarse-to-fine video generation pipeline (Ho et al., 2022a; Blattmann et al., 2023a) which starts with a module to produce keyframes at a low resolution and a low frame rate. Then we employs a temporal interpolation module and super-resolution module to increase temporal and spatial quality respectively.

In these modules, prior studies typically employ either pixel-based or latent-based VDMs across all modules. While purely pixel-based VDMs tend to have heavy computational costs, exclusively latent-based VDMs can result in poor text-video alignment and motion inconsistencies. In contrast, we combine them into MPL-Video as shown in Fig. 3. To accomplish this, we employ pixel-based VDMs for the keyframe module and the temporal interpolation module at a low resolution, producing key frames of precise text-video alignment and natural motion with low computational cost. Regarding super-resolution, we find that latent-based VDMs, despite their inaccurate text-video alignment, can be re-purposed to translate low-resolution video to high-resolution video, while maintaining the original appearance and the accurate text-video alignment of low-resolution video. Inspired by this finding, for the first time, we propose a novel two-stage super-resolution module that first employs pixel-based VDMs to upsample the video from $64 \times 40$ to $256 \times 160$ and then design a novel expert translation module based on latent-based VDMs to further upsample it to $572 \times 320$ with low computation cost.

In summary, our paper makes the following key contributions:

- Upon examining pixel and latent VDMs, we discovered that: 1) pixel VDMs excel in generating low-resolution videos with more natural motion and superior text-video synchronization compared to latent VDMs; 2) when using the low-resolution video as an initial guide, conventional latent VDMs can effectively function as super-resolution tools by **simple expert translation**, refining spatial clarity and creating high-quality videos with greater efficiency than pixel VDMs.

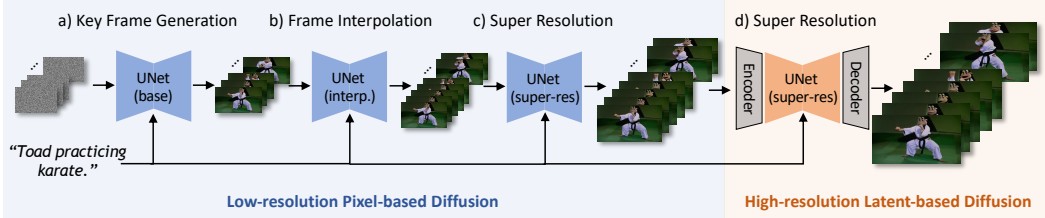

Figure 3: Overview of MPL-Video. Pixel-based VDMs produce videos of lower resolution with better text-video alignment, while latent-based VDMs upscale these low-resolution videos from pixel-based VDMs to then create high-resolution videos with low computation cost.

- We are the first to integrate the strengths of both pixel and latent VDMs, resulting into a novel video generation model that can produce high-resolution videos of precise text-video alignment at low computational cost (15G GPU memory during inference).
- Our approach achieves state-of-the-art performance on standard benchmarks including UCF-101 and MSR-VTT.

## 2 PREVIOUS WORK

**Text-to-image generation.** (Reed et al., 2016) stands as one of the initial methods that adapts the unconditional Generative Adversarial Network (GAN) introduced by (Goodfellow et al., 2014) for text-to-image (T2I) generation. Later versions of GANs delve into progressive generation, as seen in (Zhang et al., 2017) and (Hong et al., 2018). Meanwhile, works like (Xu et al., 2018) and (Zhang et al., 2021) seek to improve text-image alignment. Recently, diffusion models have contributed prominently to advancements in text-driven photorealistic and compositional image synthesis (Ramesh et al., 2022; Saharia et al., 2022). For attaining high-resolution imagery, two prevalent strategies emerge. One integrates cascaded super-resolution mechanisms within the RGB domain (Nichol et al., 2021; Ho et al., 2022b; Saharia et al., 2022; Ramesh et al., 2022). In contrast, the other harnesses decoders to delve into latent spaces (Rombach et al., 2022; Gu et al., 2022). Owing to the emergence of robust text-to-image diffusion models, we are able to utilize them as solid initialization of text to video models.

**Text-to-video generation.** Past research has utilized a range of generative models, including GANs (Vondrick et al., 2016; Saito et al., 2017; Tulyakov et al., 2018; Tian et al., 2021; Shen et al., 2023), Autoregressive models (Srivastava et al., 2015; Yan et al., 2021; Le Moing et al., 2021; Ge et al., 2022; Hong et al., 2022), and implicit neural representations (Skorokhodov et al., 2021; Yu et al., 2021). Inspired by the notable success of the diffusion model in image synthesis, several recent studies have ventured into applying diffusion models for both conditional and unconditional video synthesis (Voleti et al., 2022; Harvey et al., 2022; Zhou et al., 2022; Wu et al., 2022b; Blattmann et al., 2023b; Khachatryan et al., 2023; Höppe et al., 2022; Voleti et al., 2022; Yang et al., 2022; Nikankin et al., 2022; Luo et al., 2023; An et al., 2023; Wang et al., 2023b). Several studies have investigated the hierarchical structure, encompassing separate keyframes, interpolation, and super-resolution modules for high-fidelity video generation. Magicvideo (Zhou et al., 2022) and Video LDM (Blattmann et al., 2023a) ground their models on latent-based VDMs. On the other hand, PYoCo (Ge et al., 2023), Make-A-Video (Singer et al., 2022), Imagen Video (Ho et al., 2022a) and NUWA-XL (Yin et al., 2023) anchor their models on pixel-based VDMs. Contrary to these approaches, our method seamlessly integrates both pixel-based and latent-based VDMs.

## 3 *MPL-Video*

### 3.1 PRELIMINARIES

**Denoising Diffusion Probabilistic Models (DDPMs).** DDPMs, as detailed in (Ho et al., 2020), represent generative frameworks designed to reproduce a consistent forward Markov chain $x_1, \ldots, x_T$. Considering a data distribution $x_0 \sim q(x_0)$, the Markov transition $q(x_t|x_{t-1})$ is conceptualized as

a Gaussian distribution, characterized by a variance $\beta_t \in (0, 1)$. Formally, this is defined as:

$$q(x_t|x_{t-1}) = \mathcal{N}(x_t; \sqrt{1-\beta_t}x_{t-1}, \beta_t\mathbb{I}), \quad t = 1, \ldots, T. \tag{1}$$

Applying the principles of Bayes and the Markov characteristic, it's feasible to derive the conditional probabilities $q(x_t|x_0)$ and $q(x_{t-1}|x_t, x_0)$, represented as:

$$q(x_t|x_0) = \mathcal{N}(x_t; \sqrt{\bar{\alpha}_t}x_0, (1-\bar{\alpha}_t)\mathbb{I}), \quad t = 1, \ldots, T, \tag{2}$$

$$q(x_{t-1}|x_t, x_0) = \mathcal{N}(x_{t-1}; \tilde{\mu}_t(x_t, x_0), \tilde{\beta}_t\mathbb{I}), \quad t = 1, \ldots, T, \tag{3}$$

*where* $\alpha_t = 1 - \beta_t$, $\bar{\alpha}_t = \prod_{s=1}^t \alpha_s$, $\tilde{\beta}_t = \frac{1-\bar{\alpha}_{t-1}}{1-\bar{\alpha}_t}\beta_t$, $\tilde{\mu}_t(x_t, x_0) = \frac{\sqrt{\bar{\alpha}_t}\beta_t}{1-\bar{\alpha}_t}x_0 + \frac{\sqrt{\alpha_t}(1-\bar{\alpha}_{t-1})}{1-\bar{\alpha}_t}x_t$. In order to synthesize the chain $x_1, \ldots, x_T$, DDPMs utilize a reverse approach, characterized by a prior $p(x_T) = \mathcal{N}(x_T; 0, \mathbb{I})$ and Gaussian transitions. This relation is:

$$p_\theta(x_{t-1}|x_t) = \mathcal{N}(x_{t-1}; \mu_\theta(x_t, t), \Sigma_\theta(x_t, t)), \quad t = T, \ldots, 1. \tag{4}$$

The model's adaptable parameters $\theta$ are optimized to ensure the synthesized reverse sequence aligns with the forward sequence.

In their essence, DDPMs adhere to the variational inference strategy, focusing on enhancing the variational lower bound of the negative log-likelihood. Given the KL divergence among Gaussian distributions, this approach is practical. In practice, this framework resembles a series of weight-shared denoising autoencoders $\epsilon_\theta(x_t, t)$, trained to render a cleaner version of their respective input $x_t$. This is succinctly represented by: $\mathbb{E}_{x,\epsilon\sim\mathcal{N}(0,1),t}\left[\|\epsilon - \epsilon_\theta(x_t, t)\|_2^2\right]$.

**UNet architecture for text to image model.** The UNet model is introduced by (Spr, 2015) for biomedical image segmentation. Popular UNet for text-to-image diffusion model usually contains multiple down, middle, and up blocks. Each block consists of a resent2D layer, a self-attention layer, and a cross-attention layer. Text condition $c$ is inserted into to cross-attention layer as keys and values. For a text-guided Diffusion Model, with the text embedding $c$ the objective is given by:

$$\mathbb{E}_{x,\epsilon\sim\mathcal{N}(0,1),t,c}\left[\|\epsilon - \epsilon_\theta(x_t, t, c)\|_2^2\right]. \tag{5}$$

### 3.2 TURN IMAGE UNET TO VIDEO

We incorporate the spatial weights from a robust text-to-image model. To endow the model with temporal understanding and produce coherent frames, we integrate temporal layers within each UNet block. Specifically, after every Resnet2D block, we introduce a temporal convolution layer consisting of four 1D convolutions across the temporal dimension. Additionally, following each self and cross-attention layer, we implement a temporal attention layer to facilitate dynamic temporal data assimilation. Specifically, a frame-wise input video $x \in \mathcal{R}^{N \times C \times H \times W}$, where $C$ is the number of channels, $H$ and $W$ are the spatial latent dimensions, and $N$ is the number of frames. The spatial layers regard the video as a batch of independent images (by transposing the temporal axis into the batch dimension), and for each temporal layer, the video is reshaped back to temporal dimensions.

### 3.3 PIXEL-BASED KEYFRAME GENERATION MODEL

Given a text input, we initially produce a sequence of keyframes using a pixel-based Video UNet at a very low spatial and temporal resolution. This approach results in improved text-to-video alignment. The reason for this enhancement is that we do not require the keyframe modules to prioritize appearance clarity or temporal consistency. As a result, the keyframe modules pays more attention to the text guidance. The training objective for the keyframe modules is following Eq. 5.

**Why we choose pixel diffusion over latent diffusion here?** Latent diffusion employs an encoder to transform the original input $x$ into a latent space. This results in a reduced spatial dimension, for example, $H/8, W/8$, while concentrating the semantics and appearance into this latent domain. For generating keyframes, our objective is to have a smaller spatial dimension, like $64 \times 40$. If we opt for latent diffusion, this spatial dimension would shrink further, perhaps to around $8 \times 5$, which might not be sufficient to retain ample spatial semantics and appearance within the compacted latent space. On the other hand, pixel diffusion operates directly in the pixel domain, keeping the original spatial dimension intact. This ensures that necessary semantics and appearance information are preserved. For the following low resolution stages, we all utilize pixel-based VDMs for the same reason.

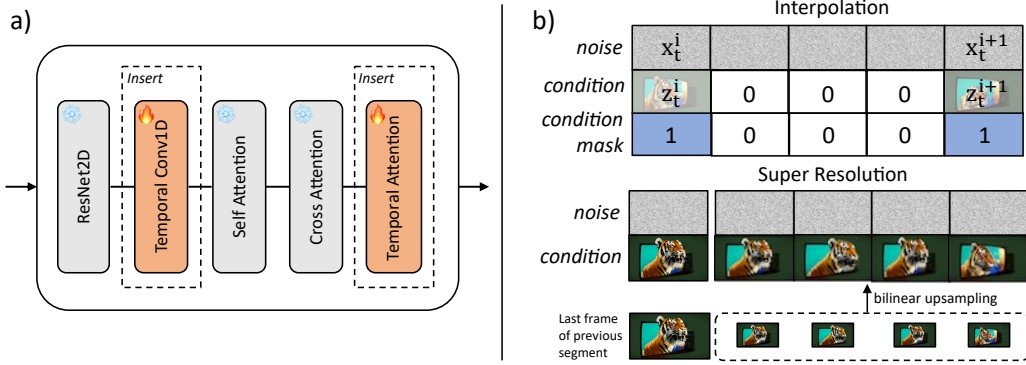

Figure 4: 3D UNet and input of the UNet of interpolation and super-resolution modules. (a) shows that how we insert temporal modules into 2D UNet. (b) explains the input for interpolation and first super-resolution UNet.

## 3.4 TEMPORAL INTERPOLATION MODEL

To enhance the temporal resolution of videos we produce, we suggest a pixel-based temporal interpolation diffusion module. This method iteratively interpolates between the frames produced by our keyframe modules. The pixel interpolation approach is built upon our keyframe modules, with all parameters fine-tuned during the training process. We employ the masking-conditioning mechanism, as highlighted in (Blattmann et al., 2023a), where the target frames for interpolation are masked. In addition to the original pixel channels $C$, as shown in Fig. 4 we integrate 4 supplementary channels into the U-Net's input: 3 channels are dedicated to the RGB masked video input, while a binary channel identifies the masked frames. As depicted in the accompanying figure, during a specific noise timestep, we interpolate three frames between two consecutive keyframes, denoted as $x_t^i$ and $x_t^{i+1}$. For the added 3 channels, values of $z_t^i$ and $z_t^{i+1}$ remain true to the original pixel values, while the interpolated frames are set to zero. For the final mask channel, the mask values $m^i$ and $m^{i+1}$ are set to 1, signifying that both the initial and concluding frames are available, with all others set to 0. In conclusion, we merge these components based on the channel dimension and input them into the U-Net. For $z_t^i$ and $z_t^{i+1}$, we implement noise conditioning augmentation. Such augmentation is pivotal in cascaded diffusion models for class-conditional generation, as observed by (Ho et al., 2022a), and also in text-to-image models as noted by (He et al., 2022). Specifically, this method aids in the simultaneous training of diverse models in the cascade. It minimizes the vulnerability to domain disparities between the output from one cascade phase and the training inputs of the following phase. Let the interpolated video frames be represented by $x^, \in \mathcal{R}^{4N \times C \times H \times W}$. Based on Eq. 5, we can formulate the updated objective as:

$$\mathbb{E}_{x^,,z,m,\epsilon \sim \mathcal{N}(0,1),t,c} \left[ \| \epsilon - \epsilon_\theta([x_t^,, z, m], t, c) \|_2^2 \right]. \tag{6}$$

## 3.5 SUPER-RESOLUTION AT LOW SPATIAL RESOLUTION

To improve the spatial quality of the videos, we introduce a pixel super-resolution approach utilizing the video UNet. For this enhanced spatial resolution, we also incorporate three additional channels, which are populated using a bilinear upscaled low-resolution video clip, denoted as $x_u^{,,} \in \mathcal{R}^{4N \times C \times 4H \times 4W}$ through bilinear upsampling. In line with the approach of (Ho et al., 2022c), we employ Gaussian noise augmentation to the upscaled low resolution video condition during its training phase, introducing a random signal-to-noise ratio. The model is also provided with this sampled ratio. During the sampling process, we opt for a consistent signal-to-noise ratio, like 1 or 2. This ensures minimal augmentation, assisting in the elimination of artifacts from the prior phase, yet retaining a significant portion of the structure.

Given that the spatial resolution remains at an upscaled version throughout the diffusion process, it's challenging to upscale all the interpolated frames, denoted as $x^, \in \mathcal{R}^{4N \times C \times H \times W}$, to $x^{,,} \in$

$\mathcal{R}^{4N \times C \times 4H \times 4W}$ simultaneously on a standard GPU with 24G memory. Consequently, we must divide the frames into four smaller segments and upscale each one individually.

However, the continuity between various segments is compromised. To rectify this, as depicted in the Fig. 4, we take the upscaled last frame of one segment to complete the three supplementary channels of the initial frame in the following segment.

## 3.6 SUPER-RESOLUTION AT HIGH SPATIAL RESOLUTION

Through our empirical observations, we discern that a latent-based VDM can be effectively utilized for enhanced super-resolution with high fidelity. Specifically, we design a distinct latent-based VDM that is tailored for high-caliber, high-resolution data. We then apply a noising-denoising procedure, as outlined by *SDEdit* (Meng et al., 2021), to the samples from the preliminary phase. As pointed out by (Balaji et al., 2022), various diffusion steps assume distinct roles during the generation process. For instance, the initial diffusion steps, such as from 1000 to 900, primarily concentrate on recovering the overall spatial structure, while subsequent steps delve into finer details. Given our success in securing well-structured low-resolution videos, we suggest adapting the latent VDM to specialize in high-resolution detail refinement. More precisely, we train a UNet for only the 0 to 900 timesteps (with 1000 being the maximum) instead of the typical full range of 0 to 1000, directing the model to be an expert emphasizing high-resolution nuances. This strategic adjustment significantly enhances the end video quality, namely expert translation. During the inference process, we use bilinear upsampling on the videos from the prior stage and then encode these videos into the latent space. Subsequently, we carry out diffusion and denoising directly in this latent space using the latent-based VDM model, while maintaining the same text input. This results in the final video, denoted as $x''' \in \mathcal{R}^{4N \times C \times 16H \times 16W}$.

**Why we choose latent-based VDM over pixel-based VDM here?** Pixel-based VDMs work directly within the pixel domain, preserving the original spatial dimensions. Handling high-resolution videos this way can be computationally expensive. In contrast, latent-based VDMs compress videos into a latent space (for example, downscaled by a factor of 8), which results in a reduced computational burden. Thus, we opt for the latent-based VDMs in this context.

## 4 EXPERIMENTS

### 4.1 IMPLEMENTATION DETAILS

For the generation of pixel-based keyframes, we utilized DeepFloyd[1] as our pre-trained Text-to-Image model for initialization, producing videos of dimensions $8 \times 64 \times 40 \times 3(N \times H \times W \times 3)$. In our interpolation model, we initialize the weights using the keyframes generation model and produce videos with dimensions of $29 \times 64 \times 40 \times 3$. For our initial model, we employ DeepFloyd's SR model for spatial weight initialization, yielding videos of size $29 \times 256 \times 160$. In the subsequent super-resolution model, we modify the ModelScope text-to-video model and use our proposed expert translation to generate videos of $29 \times 576 \times 320$.

The dataset we used for training is WebVid-10M (Bain et al., 2021). Training and hyperparameter-details can be found in appendix Table 5.

Table 1: Zero-shot text-to-video generation on UCF-101. Our approach achieves competitive results in both inception score and FVD metrics.

| Method | IS (↑) | FVD (↓) |
|---|---|---|
| CogVideo (Hong et al., 2022) (English) | 25.27 | 701.59 |
| Make-A-Video (Singer et al., 2022) | 33.00 | **367.23** |
| MagicVideo (Zhou et al., 2022) | - | 655.00 |
| Video LDM (Blattmann et al., 2023a) | 33.45 | 550.61 |
| VideoFactory (Wang et al., 2023b) | - | 410.00 |
| MPL-Video (ours) | **35.42** | 394.46 |

---

[1] https://github.com/deep-floyd/IF

Table 2: Quantitative comparison with state-of-the-art models on MSR-VTT. Our approach achieves the state-of-the-art performance.

| Models | FID-vid ($\downarrow$) | FVD ($\downarrow$) | CLIPSIM ($\uparrow$) |
|---|---|---|---|
| NÜWA (Wu et al., 2022a) | 47.68 | - | 0.2439 |
| CogVideo (Chinese) (Hong et al., 2022) | 24.78 | - | 0.2614 |
| CogVideo (English) (Hong et al., 2022) | 23.59 | 1294 | 0.2631 |
| MagicVideo (Zhou et al., 2022) | - | 1290 | - |
| Video LDM (Blattmann et al., 2023a) | - | - | 0.2929 |
| Make-A-Video (Singer et al., 2022) | 13.17 | - | 0.3049 |
| ModelScopeT2V (Wang et al., 2023a) | **11.09** | 550 | 0.2930 |
| MPL-Video (ours) | 13.08 | **538** | **0.3076** |

Table 3: Human evaluation on state-of-the-art open-sourced text-to-video models.

| | Video Quality | Text-Video alignment | Motion Fidelity |
|---|---|---|---|
| Ours *vs.* ModelScope | $62\% vs. 38\%$ | $63\% vs. 37\%$ | $63\% vs. 37\%$ |
| Ours *vs.* ZeroSope | $62\% vs. 38\%$ | $58\% vs. 42\%$ | $59\% vs. 41\%$ |

## 4.2 QUANTITATIVE RESULTS

**UCF-101 Experiment.** For our preliminary evaluations, we employ IS and FVD metrics. UCF-101 stands out as a categorized video dataset curated for action recognition tasks. When extracting samples from the text-to-video model, following PYoCo (Ge et al., 2023), we formulate a series of prompts corresponding to each class name, serving as the conditional input. This step becomes essential for class names like *jump rope*, which aren't intrinsically descriptive. We generate 20 video samples per prompt to determine the IS metric. For FVD evaluation, we adhere to methodologies presented in prior studies (Le Moing et al., 2021; Tian et al., 2021) and produce 2,048 videos.

From the data presented in Table 1, it's evident that MPL-Video's zero-shot capabilities outperform or are on par with other methods. This underscores MPL-Video's superior ability to generalize effectively, even in specialized domains. It's noteworthy that our keyframes, interpolation, and initial super-resolution models are solely trained on the publicly available WebVid-10M dataset, in contrast to the Make-A-Video models, which are trained on other data.

**MSR-VTT Experiment.** The MSR-VTT dataset (Xu et al., 2016) test subset comprises $2,990$ videos, accompanied by $59,794$ captions. Every video in this set maintains a uniform resolution of $320 \times 240$. We carry out our evaluations under a zero-shot setting, given that MPL-Video has not been trained on the MSR-VTT collection. In this analysis, MPL-Video is compared with state-of-the-art models, on performance metrics including FID-vid (Heusel et al., 2017), FVD (Unterthiner et al., 2018), and CLIPSIM (Wu et al., 2021). For FID-vid and FVD assessments, we randomly select 2,048 videos from the MSR-VTT testing division. CLIPSIM evaluations utilize all the captions from this test subset, following the approach (Singer et al., 2022). All generated videos consistently uphold a resolution of $256 \times 256$.

Table 2 shows that, MPL-Video achieves the best FVD performance (with a score of 538). This suggests a remarkable visual congruence between our generated videos and the original content. Moreover, our model secures a notable CLIPSIM score of 0.3076, emphasizing the semantic coherence between the generated videos and their corresponding prompts. It is noteworthy that our CLIPSIM score surpasses that of Make-A-Video (Singer et al., 2022), despite the latter having the benefit of using additional training data beyond WebVid-10M.

**Human evaluation.** We gather an evaluation set comprising 120 prompts that encompass camera control, natural scenery, food, animals, people, and imaginative content. The survey is conducted on Amazon Mechanical Turk. Following Make a Video (Singer et al., 2022), we assess video quality, the accuracy of text-video alignment and motion fidelity. In evaluating video quality, we present two videos in a random sequence and inquire from annotators which one possesses superior quality. When considering text-video alignment, we display the accompanying text and prompt annotators to determine which video aligns better with the given text, advising them to overlook quality concerns. For motion fidelity, we let annotators to determine which video has the most natural notion. As shown in Table 3, our method achieves the best human preferences on all evaluation parts.

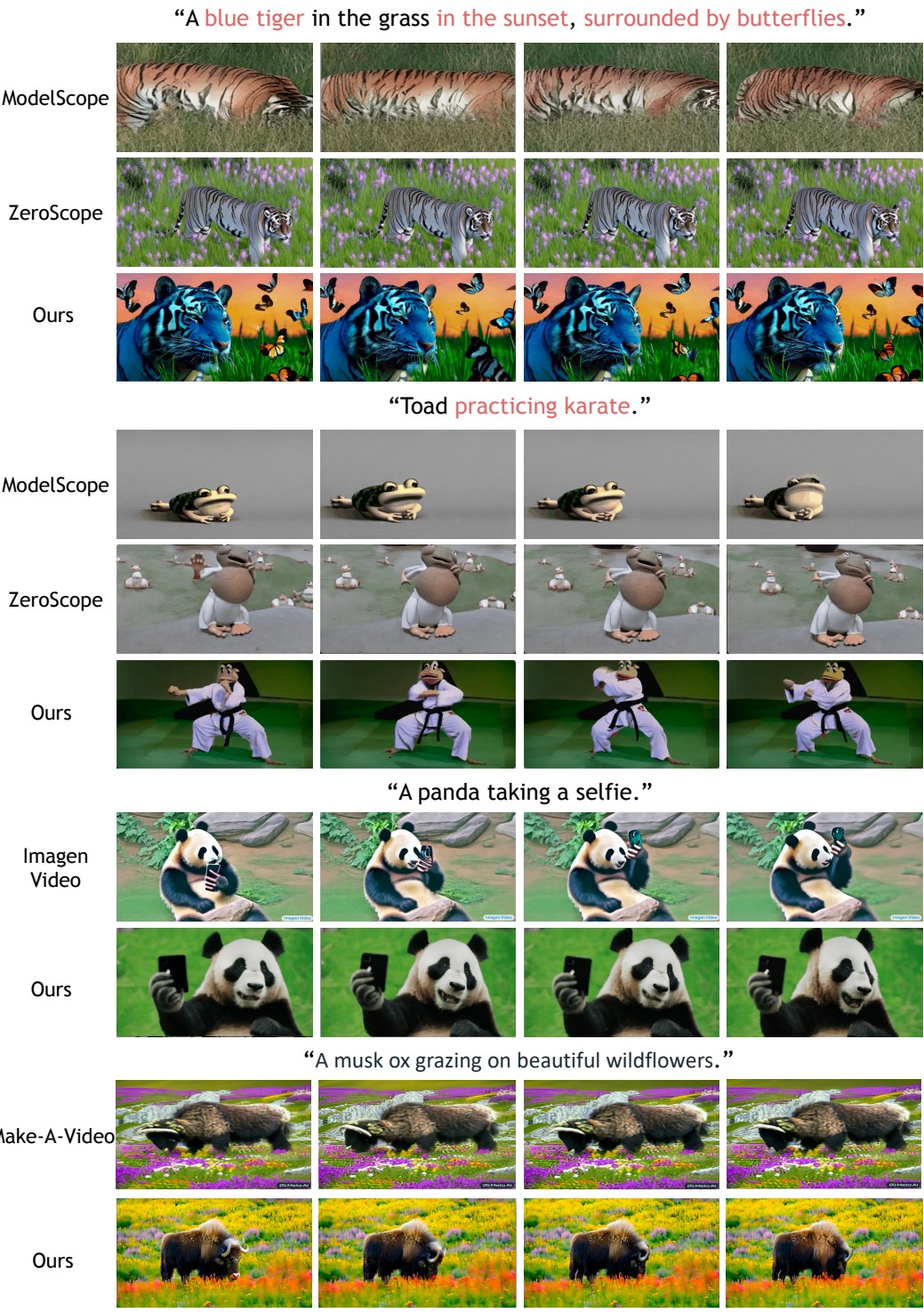

Figure 5: Qualitative comparison with existing video generative models. Words in red highlight the misalignment between text and video in other open-source approaches (*i.e.*, ModelScope and ZeroScope), whereas our method maintains proper alignment. Videos from closed-source approaches (*i.e.*, Imagen Video and Make-A-Video) are obtained from their websites.

"A married couple embraces in front of a burning house."

Figure 6: Qualitative comparison for our expert translation. With expert translation, the visual quality is significantly improved.

Table 4: Comparisons of different combinations of pixel-based and latent-based VDMs in terms of text-video similarity and memory usage during inference.

| Low resolution stage | High resolution stage | CLIPSIM | Max memory | FVD | Text-Video Alignment | Visual Quality |
|---|---|---|---|---|---|---|
| latent-based | latent-based | 0.2934 | 15GB | 584 | 23% | 38% |
| latent-based | pixel-based | – | 72GB | – | – | – |
| pixel-based | pixel-based | – | 72GB | – | – | – |
| pixel-based | latent-based | **0.3072** | 15GB | 542 | 77% | 62% |

## 4.3 QUALITATIVE RESULTS

As depicted in Fig. 5, our approach exhibits superior text-video alignment and visual fidelity compared to the recently open-sourced ModelScope (Wang et al., 2023a) and ZeroScope[2]. Additionally, our method matches or even surpasses the visual quality of the current state-of-the-art methods, including Imagen Video and Make-A-Video.

## 4.4 ABLATION STUDIES.

**Impact of different combinations of pixel-based and latent-based VDMs.** To assess the integration method of pixel and latent-based VDMs, we conduct several ablations. For fair comparison, we employe the T5 encoder (Raffel et al., 2020) for text embedding in all low-resolution stages and the CLIP text encoder (Radford et al., 2021) for high-resolution stages. As indicated in Tab. 4, utilizing pixel-based VDMs in the low-resolution stage and latent diffusion for high-resolution upscaling results in the highest CLIP score with reduced computational expenses. On the other hand, implementing pixel-based VDMs during the high-resolution upscaling stage demands significant computational resources. These findings reinforce our proposition that combining pixel-based VDMs in the low-resolution phase and latent-based VDMs in the high-resolution phase can enhance text-video alignment and visual quality while minimizing computational costs.

**Impact of expert translation of latent-based VDM as super-resolution model.** We provide visual comparison between models with and without expert translation. As elaborated in Section 3.6, "with expert translation" refers to training the latent-based VDMs using timesteps 0-900 (with a maximum timestep of 1000), while "w/o expert translation" involves standard training with timesteps 0-1000. As evident in Fig. 6, the model with expert translation produces videos of superior visual quality, exhibiting fewer artifacts and capturing more intricate details.

## 5 CONCLUSION

We introduce MPL-Video, an innovative model that marries the strengths of pixel and latent based VDMS. Our approach employs pixel-based VDMs for initial video generation, ensuring precise text-video alignment and motion portrayal, and then uses latent-based VDMs for super-resolution, transitioning from a lower to a higher resolution efficiently. This combined strategy offers high-quality text-to-video outputs while optimizing computational costs.

---

[2]https://huggingface.co/cerspense/zeroscope-v2-576w

## 6 Ethics Statement

Our pretrained T2I model, Deep-IF, is trained using web data, and our models utilize WebVid-10M. Given this, there's a potential for our method to not only learn but also amplify societal biases, which could include inappropriate or NSFW content. To address this, we can integrate the CLIP model to detect NSFW content and filter out such instances.

## 7 Reproducibility Statement

We take the following steps to guarantee reproducibility: (1) Our codes, along with model weights, will be public available. (2) The training and hyperparameter details can be found in appendix Table 5.

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

# A Appendix

## A.1 Discussion and limitations.

The present method allows for the creation of videos lasting 4 seconds with one scene. The capability to produce extended videos that encompass multiple scenes and events, portraying more intricate narratives, remains a topic for future exploration. Furthermore, the absence of established benchmarks for large-scale generative models poses challenges in comparing different works and tracking advancements. Our ongoing efforts are directed towards enhancing benchmarks for these generative models.

Table 5: Hyperparameters for our all models presented.

| Hyperparameter | Keyframe Module | Interpolation Module | First Superresolution | Second Superresolution |
|---|---|---|---|---|
| Space | pixel | pixel | pixel | latent |
| $fps$ | 2 | 8 | 8 | 8 |
| Channels | 320 | 320 | 128 | 320 |
| Depth | 4 | 4 | 5 | 4 |
| Channel multiplier | 1,2, 4,4 | 1,2,4,4 | 1,2,4,6,6 | 1,2,4,4 |
| Head channels | 64 | 64 | 64 | 64 |
| *Training* | | | | |
| Parameterization | $\varepsilon$ | $\varepsilon$ | $\mathbf{v}$ | $\varepsilon$ |
| # train steps | 120K | 40K | 40K | 120K |
| Learning rate | $10^{-4}$ | $10^{-4}$ | $10^{-4}$ | $10^{-4}$ |
| Batch size per GPU | 1 | 2 | 1 | 1 |
| # GPUs | 48 | 16 | 16 | 24 |
| GPU-type | A100-40GB | A100-40GB | A100-40GB | A100-40GB |
| $p_{\text{drop}}$ | 0.1 | 0.1 | 0.1 | 0.1 |
| **Diffusion Setup** | | | | |
| Diffusion steps | 1000 | 1000 | 1000 | 1000 |
| Noise schedule | Linear | Linear | Linear | Linear |
| $\beta_0$ | $10^{-4}$ | $10^{-4}$ | $10^{-4}$ | 0.0015 |
| $\beta_T$ | 0.02 | 0.02 | 0.02 | 0.0195 |
| **Sampling Parameters** | | | | |
| Sampler | DPM++ | DPM++ | DPM++ | DDIM |
| Steps | 75 | 50 | 125 | 40 |
| $\eta$ | 1.0 | 1.0 | 1.0 | 1.0 |

## A.2 Training and Hyperparameter details.

We list details of our models in Table 5.

## A.3 More Video Results.

For the convenience, we include more video results on an anonymous webpage https://anonymous-iclr-1864.github.io/

