# OpenReview forum: "Marrying Pixel and Latent Diffusion Models for Text-to-Video Generation"
_ICLR.cc/2024/Conference — ICLR 2024 Conference Withdrawn Submission_

### Official Review · Reviewer_554b · 2023-10-28

**Soundness:** 2 fair
**Presentation:** 3 good
**Contribution:** 3 good
**Rating:** 5
**Confidence:** 4

**Summary:**

This paper introduces MPL-Video, a hybrid model for text-to-video generation that combines pixel-based and latent-based Video Diffusion Models (VDMs). MPL-Video overcomes limitations of previous methods, e.g., precise text-to-video alignment, by initially employing pixel-based VDMs to generate a low-resolution video with strong text-video correlation. Subsequently, a uniquely proposed expert translation method leveraging latent-based VDMs is applied to enhance the low-resolution output to high resolution. MPL-Video demonstrates superior performance compared to its pixel-based and latent-based counterparts and has significantly lower GPU memory usage during inference (15G as opposed to 72G).

**Strengths:**

1. The paper introduces an approach by combining pixel-based and latent-based VDMs, overcoming the individual limitations of each. This hybrid model, MPL-Video, employs pixel-based VDMs for keyframe modules and temporal interpolation at low resolution, ensuring precise text-video alignment and natural motion with reduced computational costs.

2. The proposed method ensures enhanced super-resolution of videos, achieving precise text-to-video alignments with optimized computational costs, thereby advancing the field by offering a balanced solution that does not compromise quality for computational efficiency.

**Weaknesses:**

1. The assertion that
> Pixel-based VDMs can generate motion accurately aligned with the textual prompt

appears somewhat unsubstantiated. The manuscript could be fortified by providing more compelling empirical evidence or a thorough rationale that unequivocally corroborates this claim, thereby enhancing the reliability of the proposed methodology.

2.  Section 3.3 deliberates the choice of pixel diffusion over latent diffusion. However, the justification proffered seems to lack cogency. The methodology initiates with a smaller resolution in the pixel space, transitioning later to a higher resolution in the latent space. A more elaborate elucidation regarding this choice would be instrumental. Specifically, it would be insightful to comprehend why a high resolution is not directly employed in conjunction with latent diffusion, elucidating the inherent advantages of beginning with a lower resolution in the pixel space.

3. The manuscript primarily focuses on comparing GPU memory savings as a metric for evaluation. While this is undoubtedly a vital aspect, a more holistic approach considering additional pivotal parameters such as inference latency would substantially enrich the comparative analysis.

**Questions:**

1.Could the authors please clarify whether the visualization results presented in Figure 2 were generated using identical random seeds and training data? Based on prior experience, utilizing consistent random seeds and training datasets typically results in latent-based VDMs exhibiting similar patterns at both low and high resolutions. However, the patterns depicted in Figure 2 seem to considerably diverge from this expectation. A clarification regarding this discrepancy would be highly appreciated.

---

> ### Author Response · Authors · 2023-11-12
> **Response to Reviewer 554b [1/2]**
>
> Thanks for your insightful review! We have prepared a comprehensive response and hope it satisfactorily addresses your concerns.
> ### **Q1:The claim is somewhat unsubstantiated.**
> ### **A1.**
> Thanks for your valuable question. "Imagen Video" and "Make a Video" have shown that using hierarchical pixel VDMs to generate the videos in very low resolutions (e.g., 64x40)  first and then upscale videos to higher resolutions (576x320) can lead to superior results. We make our claim based on previous SOTA methods.
>  Moreover, to further validate the claim that pixel VDM can generate motion accurately aligned with the textual prompt, we conduct the clip-text sore and  human evaluation(details can be found in paper experiment section) between pixel VDMs with 64x40 resolution and latent VDMs with 256x160 resolution.) and tell the annotator only to consider the text-video alignment and motion fidelity regardless of visual quality. As shown in Tab. A, pixel VDMs achieve better results, which demonstrates our claim.
>
> |                  | Clip-text score | Text-Video alignment | Motion Fidelity |
> |--------------|-----------------|------------------|------------------
> | Pixel    | 0.3031          | 72%              |    68%             |
> |Latent     | 0.2892          | 28%              |    32%             |
> ### Table A   Results using 100 complex prompts
>
>
> ### **Q2. Why a high resolution is not directly employed in conjunction with latent diffusion, elucidating the inherent advantages of beginning with a lower resolution in the pixel space?**
>
> ### **A2.**
> **First,** "Imagen Video" and "Make a Video" have shown that using hierarchical pixel VDMs to generate the videos in very low resolutions (e.g., 64x40)  first and then upscale videos to higher resolutions (576x320) can lead to superior results.
> **Second,** latent models like zeroscope, VideoLDM, and modelscope can directly generate high-resolution videos. However, generating directly in high resolution causes these models to focus more on spatial appearance rather than text-video alignment, leading to weaker alignment than seen in "Imagen Video" and "Make a Video." For example, models starting from a very low resolution can easily generate videos with text  (e.g., "ICLR" as shown in Fig. 1 of  paper), but it's much more difficult for high-resolution latent VDMs to produce such videos with text directly. As shown in Tab.B, the results  demonstrate that generating a very low-resolution video first and then applying super-resolution results in much better video-text alignment than starting with high-resolution generation using a latent model.  Furthermore, as shown in Tab.C,  as the resolution of the latent model increases, the video-text alignment is weaker, proving that the model increasingly focuses on spatial appearance over text alignment with higher resolution. Our paper's Tab. 4 confirms that combining pixel models at a low resolution stage with latent models at a high resolution stage is the most effective combination.  All above results valid the benefit of beginning with a lower resolution in the pixel space.
>
> |                  | Clip-text score | Human preference |
> |--------------|-----------------|------------------|
> | Low -> High      | 0.3026          | 73%              |
> | High -> High     | 0.2874          | 27%              |
> ### Table B     Video-text alignment results using 100 complex prompts
>
> | Resolution | Clip-text score | Human preference |
> |------------|-----------------|-----------------|
> | 256x160    | 0.2926          | 62%              |
> | 512x320    | 0.2874          | 38%              |
> ### Table C Impact of resolution of latent VDMs for Video-text alignment.
>
> ### **Q3. Inference latency**
>
> ### **A3.**
> We replicated the Make-A-Video model by precisely matching its parameters and network architecture for inference time comparisons. As shown in Tab.D, the results show that our method is faster and more memory efficient  than the previous SOTA method, Make-a-Video.
> | Stage                     | Prior        | Keyframes        | Temporal Interpolation | Superresolution1 | Final Superresolution | Total                   |
> |---------------------------|--------------|------------------|------------------------|------------------|-----------------------|-------------------------|
> | Step (Make/ Ours)         | 64/–         | 100/75           | 50/75                  | 50/50            | 50/40                 |  –                 |
> | Para. (Make/ Ours)    | 1.3B/–       | 3.1B/1.7B        | 3.1B/1.7B              | 1.4B/0.8B        | 0.7B/1.8B             | 9.6B/6B                   |
> | Make (s)                  | 3s           | 58s              | 62s                    | 70s              | 63s                   | 256s (52Gb)             |
> | Ours (s)                  | –            | 30s              | 60s                    | 65s              | 23s                   | 178s (15GB)             |
> ### Table D  Inference comparisons with Make-A-Video.

---

> > ### Author Response · Authors · 2023-11-12
> > **Response to Reviewer 554b[2/2]**
> >
> > ### **Q4. Random seeds and  diverge.**
> >
> > ### **A4.**
> > Thank you for highlighting this. Firstly, it's important to note that the seeds used for visualization are chosen randomly. Our claim can be evidenced by the qualitative results displayed in A1 and A2. Secondly, due to the distinct convergence speeds and training curves between latent and pixel spaces, which are further influenced by resolution, the same seed can lead to differing appearances even at identical steps of training.

---

### Official Review · Reviewer_WLJK · 2023-10-28

**Soundness:** 3 good
**Presentation:** 3 good
**Contribution:** 2 fair
**Rating:** 5
**Confidence:** 4

**Summary:**

This paper proposes to combine the pixel-based and latent-based diffusion models under the cascaded framework for text-to-video generation. They use three pixel-based modules to generate videos of size 29 × 256 × 160 and an additional latent-based module to generate videos of size 29 × 576 × 320. The motivation is that the pixel-based models provide better text-video alignment and the latent-based models require less computation and memory.

**Strengths:**

The idea of marrying pixel-based and latent-based diffusion models is clearly presented and relatively novel. The paper is generally well-written and easy to follow. The resulting model achieves competitive quantitative results on standard benchmarks and only requires 15GB memory during the inference. The qualitative results shown on the anonymous website are appealing and promising. The authors promise to release the code and models.

**Weaknesses:**

1. Motivation: I found that the argument that the latent-based model produces less text-aligned results than the pixel-based model is not very convincing since we do observe a similar level of text-video alignment in Stable Diffusion or Video LDM compared with other pixel-based diffusion models. If the reason is that the latent is too small (8×5 for 64×40 videos), then could the authors explain what might be the benefits of generating such a low-res video in the first stage? Also, one can always use a smaller compression rate if the latent needs to be larger, since a larger compression rate is only needed to reduce more computation and memory.

2. Motivation: the other motivation of the proposed method is that the latent-based model can save memory when upsampling in the high-resolution. However, pixel-based models like Make-A-Video often mitigate this issue by reducing the number of parameters in the latter upsampling modules. Thia makes sense as the upsampling task is much simpler than the generation task. Therefore, this motivation is also less convincing to me. Specifically, I wonder how is the 72GB memory calculated in Table 4.

3. Visual result: in the comparison results with Make-A-Video in Figure 5, it seems that the video generated by MPL-video has much less motion.

4. There are several unclear details in the paper. Please see the questions below.

**Questions:**

1. In Section 3.2, how does the temporal attention work exactly? Are the spatial dimensions transposed to the batch dimension, or is the attention applied to both the spatial and temporal dimensions?

2. In Section 3.5, when generating the first segment, what is used as the conditioned high-resolution frame?

3. In Section 4.1, are ModelScope model weights used to initialize the last super-resolution model?

4. The WebVid-10M dataset is known to have plenty of watermarks. Why is this not seen in the generation results?

5. In Table 4, is there a typo in the last row, which should be pixel-based for the low-res stage and latent-based for the high-res stage? If that is the case, why is the CLIPSIM score different from the one in Table 2?

---

> ### Author Response · Authors · 2023-11-12
> **Response to Reviewer WLJK [1/2]**
>
> Many thanks for your valuable feedbacks!
> ### **Q1 (1) Benefit of generating a low-res video in the first stage. (2) Small compress ratio using latent diffusion?**
>
> ### **A1.**
> **(1)** Thanks for your good points. First, Imagen Video and Make a Video have shown that using hierarchical pixel VDMs to generate the videos in very low resolutions (e.g., 64x40)  first and then upscale videos to higher resolutions (576x320) can lead to superior results. Second, latent models like zeroscope, VideoLDM, and modelscope can directly generate high-resolution videos. However, generating directly in high resolution causes these models to focus more on spatial appearance rather than text-video alignment, leading to weaker alignment than seen in "Imagen Video" and "Make a Video." For example, models starting from a very low resolution can easily generate videos with text  (e.g., "ICLR" as shown in Fig.1 of paper), but it's much more difficult for high-resolution latent VDMs to produce such videos with text directly. As shown in Tab.A, the results  demonstrate that generating a very low-resolution video first and then applying super-resolution results in much better video-text alignment than starting with high-resolution generation using a latent model. Furthermore, as shown in Tab.B,  as the resolution of the latent model increases, the video-text alignment is weaker, proving that the model increasingly focuses on spatial appearance over text alignment with higher resolution.
>
> The previous SOTA methods and above illustration of experiment results demonstrate benefits of generating such a low-res video in the first stage.
>
>
> |             | Clip-text score | Human preference |
> |------|-----|-------|
> | Low -> High      | 0.3026          | 73%              |
> | High -> High     | 0.2874          | 27%              |
> ### Table A Video-text alignment results using 100 complex prompts
>
> | Resolution | Clip-text score | Human preference |
> |-----|------|--------|
> | 256x160    | 0.2926          | 62%              |
> | 512x320    | 0.2874          | 38%              |
> ### Table B   Impact of resolution of latent VDMs for Video-text alignment.
>
> **(2)** Thanks for your insightful questions. This can be explained by the goal of latent diffusion. The primary aim of latent diffusion, as mentioned in [1], is to significantly reduce computation and memory usage. For example, stable diffusion effectively compresses a 512x512 image to a 64x64 latent size, which is an 8-fold reduction. However, when the compression ratio is minimal, like 2 times, the efficiency and training expenses are quite similar to those of pixel diffusion, also as indicated in [1]. Therefore, if the compression ratio is small, using latent diffusion might not be necessary, especially considering that latent diffusion requires training an additional autoencoder, whereas pixel diffusion does not.
>
> [1] High-Resolution Image Synthesis with Latent Diffusion Models. CVPR2022
>
> ### **Q2. (1) Motivation of using  latent diffusion for upsampling. (2) How to calucate tht 75G?**
> ### **A2.**
>
> **(1)** Thanks for your valuable questions. Although Make-A-Video has reduced the parameter count of its final superresolution model to 0.7B, it still demands significant computational resources due to its reliance on pixel-based VDM. In this model, the Unet processes a 576x320 input to predict a 572x320 noise pattern. In contrast, our latent-based VDMs handle a much smaller 72 x 40 input and predict the corresponding noise, resulting in a substantial efficiency gap. To further examine computational costs, we meticulously replicated Make-A-Video, matching its original parameters and network architecture. As Tab.C illustrates, even with 0.7B parameters, Make-A-Video's superresolution phase remains computationally intensive.
>
> Additionally, upsampling synthetic videos presents notable challenges. Since these models are trained on real data but tested on synthetic output from earlier stages, they encounter a domain gap. Moreover, ensuring temporal consistency, minimizing artifacts, and maintaining high visual quality **demands significant model complexity, which may not be feasible with reduced parameters**. Our findings, corroborated by [1], indicate that latent diffusion,  with an equal parameter count, is far more efficient than pixel-based VDMs for high-resolution image/videos upsampling and can deliver superior performance. Consequently, we opted for a latent diffusion model to upscale our high-resolution videos.
>
> | Parameter  | 0.7B | 1.8B |
> |---|---|--|
> | Max memory | 52G  | 15G  |
> ### Table C. Memory cost of the final superresolution model
>
> **(2)** The maximum memory usage is primarily due to the final superresolution stage. Initially, we implemented a pixel super resolution model with the same number of parameters as the latent superresolution model. However, this approach led to memory exhaustion. Consequently, we reduced the parameter count to 1.1B, which resulted in a memory usage of 75GB.

---

> ### Author Response · Authors · 2023-11-12
> **Response to Reviewer WLJK [2/2]**
>
> ### **Q3. Visual result: in the comparison results with Make-A-Video in Figure 5, it seems that the video generated by MPL-video has much less motion.**
> ### **A3.**
> Thanks for your  findings! It’s  just a single case amd we conduct human evaluation between Make-A-Video and our method, using the prompts and videos in Make-A-Video project pages. As shown in Tab.D, our method achieves better results.
>
> |                     | Text-Video Alignment | Visual Quality | Motion Fidelity |
> |---------------------|----------------------|----------------|-----------------|
> | Make-A-Video        | 49%                  | 35%            | 41%             |
> | Ours                | 51%                  | 65%            | 59%             |
> ### Table D. Comparisons with Make-A-Video.
>
> ### **Q4. How does temporal attention work exactly?**
> ### **A4**.
> Spatial dimensions are  transposed to the batch dimension and attention is applied to only temporal dimensions.
>
> ### **Q5. In Section 3.5, when generating the first segment, what is used as the conditioned high-resolution frame?**
>
> ### **A5**.
> We can use any image super resolution model to upsample  the first frame of the first segment as the conditioned high-resolution frame.
>
> ### **Q6. Are ModelScope model weights used to initialize the last super-resolution model?**
> ### **A6.**  Yes.
>
> ### **Q7. WebVid-10M  watermarks.**
>
> ### **A7.**
> For all quantitative results, we only use WebVid-10M  data. For final visual results, we finetune the model on around 1200 videos from InternVideo dataset. which is watermark free, to eliminate the watermark.
>
> ### **Q8. In Table 4, is there a typo in the last row, which should be pixel-based for the low-res stage and latent-based for the high-res stage? If that is the case, why is the CLIPSIM score different from the one in Table 2.**
> ### **A8.**
> Thanks for pointing it out. We correct the typo in the revised paper. And for ablation study, we train the model with 80K  steps for ablations. In table 2, the keyframes  model is trained with 100K steps. Therefore, the result has a slight difference.

---

### Official Review · Reviewer_JANw · 2023-10-29

**Soundness:** 3 good
**Presentation:** 3 good
**Contribution:** 3 good
**Rating:** 5
**Confidence:** 5

**Summary:**

This paper focus on text-to-video generation by exploring the combination of pixel-based and latent-based video diffusion models (VDMs). The authors found that pixel-based VDMs can generate accurate motion described by the text but are expensive. Latent-based VDMs are more resource-efficient but cannot cover rich visual semantic details. Therefore, the authors propose a hybrid model, which generates low-resolution videos of strong text-video correlation with pixel-based VDMs and upsamples them to high resolution with latent-based VDMs.

**Strengths:**

The motivation of mixing these two models is also intuitive. The authors conduct a comprehensive comparison between pixel-based and latent-based VDMs. They also explained how they choose between pixel diffusion and latent diffusion in different modules in their method part.

Experiments demonstrate the superiority of their proposed method in terms of its memory cost and performance.

**Weaknesses:**

1. Since there are three models involved in inference (pixel-based keyframe generation model, temporal interpolation model and 2 super-resolution models), the time cost of this method could be high.

2. The overall pipeline is not something new. It first generates keyframes and then temporally interpolates between keyframes and upsamples to higher spatial resolution. This pipeline has been proposed in previous approaches, such as make-a-video and Imagen Video and NUWA-XL[1]. The authors should also cite [1] as a related work.

3. Typos:

* $T$ is defined as the number of diffusion steps in 3.1 and it is also defined as the number of frames in 3.2.

* From my understanding, $\{z^{i}_t|t=1…T\}$ are equal. If that is true, $z^{i}$ and $z^{i+1}$ in sec. 3.4 should keep consistent with $z^{i}_t$ and $z^{i+1}_t$ in Figure 4 (b).

* C in Section 3.4 should be in the math format $C$

* “.,” in formula (6)

* In Table 2. “11.09” should be bolded instead of “13.17”, which means MPL-Video does not achieve the best performance in FID-vid as claimed in Sec. 4.2.

[1] Yin et al. NUWA-XL: Diffusion over Diffusion for eXtremely Long Video Generation. ACL 2023

**Questions:**

1. In Table 4, how different combinations of pixel-based and latent-based VDMs perform in terms of video fidelity?

2. How does the time cost of your proposed method compare to that of previous methods?

---

> ### Author Response · Authors · 2023-11-12
>
> Many thanks for your review! We have prepared a comprehensive response and hope it satisfactorily addresses your concerns.
>
> ### **Q1. Hierarchical structures(three models) are time consuming.**
> ### **A1.**
> Thanks for pointing it out.  Although hierarchical structures require more inference time compared to single-stage models, their outcomes are significantly superior, as evidenced by advanced generation methods like Imagen Video, Make-A-Video, and VideoLDMs. These state-of-the-art methods all employ hierarchical frameworks, including keyframe generation, temporal interpolation, and superresolution, for video creation. We replicated the Make-A-Video model by precisely matching its parameters and network architecture for inference time comparisons. As shown in below Tab.A, the results show that our method is faster and more memory efficient  than the previous SOTA method, Make-A-Video.
>
> | Stage                     | Prior        | Keyframes        | Temporal Interpolation | Superresolution1 | Final Superresolution | Total                   |
> |---------------------------|--------------|------------------|------------------------|------------------|-----------------------|-------------------------|
> | Step (Make/ Ours)         | 64/–         | 100/75           | 50/75                  | 50/50            | 50/40                 | --                |
> | Para. (Make/ Ours)    | 1.3B/–       | 3.1B/1.7B        | 3.1B/1.7B              | 1.4B/0.8B        | 0.7B/1.8B             | 9.6B/6B                   |
> | Make (s)                  | 3s           | 58s              | 62s                    | 70s              | 63s                   | 256s (52GB)             |
> | Ours (s)                  | –            | 30s              | 60s                    | 65s              | 23s                   | 178s (15GB)             |
> ### Table A  Inference time comparisons with Make-A-Video
>
>
> ### **Q2.  (1) Overall pipeline is not something new. (2) Missing reference.**
> ### **A2.**
> **(1)** We respectfully disagree with your opinions. **i)** The related work section highlights that Make-A-Video and Imagen Video are exclusively pixel-based Video Diffusion Models (VDMs). These models typically incur high computational costs in terms of time and GPU memory, especially for generating high-resolution videos, as the UNet diffuses noise at the video's original resolution. **In contrast to purely pixel-based structures**, our approach combines pixel-based VDMs with latent-based VDMs. This hybrid method initially generates low-resolution, text-aligned videos using pixel-based VDMs, then upscales them efficiently using latent-based VDMs. Since latent VDMs can condense high resolution videos, like 576x320, into a more compact latent space, such as 72x40, they significantly lower computational costs, as shown in Tab. A. Additionally, unlike Make-A-Video and Imagen Video, which are closed-source, we will definitely public our code and weights to contribute to the research community.
>
> **ii)** Our  final superresolution  method is distinct from the methods used in Make-A-Video and Imagen Video. In these models, three additional channels are concatenated to the UNet's input, filled with a bilinearly upscaled low-resolution video clip as a condition. In contrast, our method bilinearly upscales the low-resolution video clip and then feeds it into a VAE encoder. Subsequently, we apply DDPM noise to the encoded latent and directly denoise this noisy latent for superresolution. Essentially, we employ a video-to-video translation function for the final superresolution process, which is different from and much faster than the final superresolution of Make-A-Video  as indicated in Tab. A.
>
> **(2)** Thanks for pointing it out. NUWA-XL is a latent-based VDM and we added it into the related works  section of the revised paper.
>
> ### **Q3. Typos.**
> ### **A3.**
> Thanks for noting! We corrected them in our revised paper
>
> ### **Q4. In terms of video fidelity?**
> ### **A4**
> We add the FVD results on MSRVTT in Tab. 4 of the revised paper. Marrying pixel VDMS  in the low resolution stages and  latent VDMs in high resolution achieves the best result.
>
> ### **Q5. Inference speed compared to previous method**
> ### **A5.**
> Please refer Q1-A1.

---

### Official Review · Reviewer_uFUa · 2023-11-01

**Soundness:** 2 fair
**Presentation:** 3 good
**Contribution:** 2 fair
**Rating:** 5
**Confidence:** 5

**Summary:**

This paper proposed a new text-to-video generation model based on cascaded diffusion models, where the first text-to-video model, the temporal interpolation, and low-res SR models are based on pixel diffusion models. On the contrary, the last high-res SR model is based on the latent diffusion model, which is the main difference between the proposed method with existing baselines such as the Make-A-Video and Imagen Video, which are fully based on pixel diffusion models. Experimental results on UCF101 and MSRVTT show that the proposed method has comparable video quality while the text-following ability is improved.

**Strengths:**

- [writing] This paper is easy to follow for me, and the overall writing is very good.
- [method] The idea of combining the pixel space diffusion model and the latent space model to have both the benefits of good text following and computation saving is interesting.
- [experiment] The experimental results show that the proposed method has comparable or better video generation quality in terms of the FVD and CLIPSIM while the memory spend has been reduced.

**Weaknesses:**

- [writing] Should the last row of Table 4 be pixel-based then latent-based? Typos: "a expert".
- [method] The main weakness of the proposed method is that it is very similar to existing works like Make-A-Video in the algorithmic design, where the main difference lies in the choice of latent- or pixel-based diffusion models for different functions. Existing works (like those compared in the experiment section) have already demonstrated that both pixel and latent diffusion models can work well on text-to-video generation. I am unsure whether the combinational use can bring new insight into the community.
- [method] In the expert translation part, the authors mentioned that training the latent diffusion model only on 0-900 time steps instead of 0-1000 yields better video quality. Is there any quantitative evidence about the quality improvement?
- [method] I find the CLIPSIM improvement of the proposed method is insignificant while one of the main benefits of using the pixel-space diffusion model is that it can preserve better semantics and appearances. Are there any other metrics that can demonstrate the benefit of using a pixel-space diffusion model over latent-space ones?
- [experiment] I noticed that the Imagen Video and Make-A-Video's papers mentioned that their models have a parameter size of 16.3 and 9.7 billion, respectively. I am unsure how large is the model size of the proposed method. How to compare the above parameter size with the the max memory shown in Table 4?

**Questions:**

Please see the weakness part.

---

> ### Author Response · Authors · 2023-11-12
>
> Thank you very much for your review! We have prepared a comprehensive response and hope it satisfactorily addresses your concerns.
> ### **Q1. Typo**
> ### **A1.**
> Thanks for noting. Yes, we have corrected them in our revision.
>
> ### **Q2. Combining pixel and latent brings new insights into the community?**
> ### **A2.**
> (1)"Imagen Video" and "Make a Video" have shown that using hierarchical pixel VDMs to generate the videos in very low resolutions (e.g., 64x40)  first and then upscale videos to higher resolutions (576x320) can lead to superior results. However, pixel-based VDMs denoise through a U-Net at the original resolution, which is computationally expensive during the super-resolution process. This can be addressed by replacing the super-resolution pixel model with a latent model, significantly reducing computation costs. **This is new insight for original pixel VDMs.**
>
> (2) Latent models like zeroscope, VideoLDM, and modelscope can directly generate high-resolution videos. However, generating directly in high resolution causes these models to focus more on spatial appearance rather than text-to-video alignment, leading to weaker alignment than seen in "Imagen Video" and "Make a Video." For example, models starting from a very low resolution can easily generate videos with text  (e.g., "ICLR" as shown in Fig. 1 of  paper), but it's much more difficult for high-resolution latent VDMs to produce such videos with text directly. As shown in Tab.A, the results  demonstrate that generating a very low-resolution video first and then applying super-resolution results in much better video-text alignment than starting with high-resolution generation using a latent model. Furthermore, as shown in Tab.B,  as the resolution of the latent model increases, the video-text alignment is weaker, proving that the model increasingly focuses on spatial appearance over text alignment with higher resolution. Our paper's Tab. 4 confirms that combining pixel models at the low resolution stages with latent models at the high resolution stages is the most effective combination. **This is new insight for original latent VDMs.**
>
> **Importantly, while "Imagen Video" and "Make a Video" are closed-source, we will definitely open-source our code to benefit the community.**
>
>
> |                  | Clip-text score | Human preference |
> |------------------|-----------------|------------------|
> | Low -> High      | 0.3026          | 73%              |
> | High -> High     | 0.2874          | 27%              |
> ### Table A     Video-text alignment results using 100 complex prompts
>
>
>
>
>
> | Resolution | Clip-text score | Human preference |
> |------------|-----------------|------------------|
> | 256x160   | 0.2926          | 62%              |
> | 512x320   | 0.2874          | 38%              |
>
> ### Table B  Impact of resolution of latent VDMs for Video-text alignment.
>
>
> ### **Q3. Quantitative evidence about the quality improvement for expert translation.**
> ### **A3.**
> As shown in Tab.C, with expert translation, the model achieves better FVD on MSRVTT and better human preference on 100 complex prompts.
> |                      | FVD            | Human preference |
> |------------------|-----------------|------------------|
> | w/expert       |     538         | 86%              |
> | w/o expert    |      594            | 14%           |
> ### Table C     Impact of expert translation.
>
>
> ### **Q4. Other metrics for text-video alignments.**
>
> ### **A4.**
>  Actually, 0.01 is not a small improvement. For example, as shown in Fig.1 of the paper, “A panda beside the waterfall is holding a sign that says ‘ICLR’.", the clip score of this video is 0.3004. If a generated video contains only a panda and  the waterfall without any text about ”ICLR”, the clip score is 0.2935. To eliminate the effect of the subtle change of clip core, we add an additional human evaluation about video-text alignment in Tab.4 of the paper.
>
> ### **Q5. Parameter and GFLOPS.**
>
> ### **A5.**
> Our total number of parameters of diffusion networks is 6.0 billion.  The max memory is brought by the final superresolution stage. We also implement  the “Make a Video” architecture and the max memory is also brought by the final superresolution. Our method requires 15G  and "Make a Video" requires 52G memory to do inference.